

# Migraine aura, a predictor of near-death experiences in a crowdsourced study

Daniel Kondziella[1,2], Markus Harboe Olsen[3], Coline L. Lemale[5,6] and
Jens P. Dreier[4,5,6,7,8]

[1] Department of Neurology, Rigshospitalet, Copenhagen University Hospital, Copenhagen,
Denmark
[2] Faculty of Health and Medical Sciences, University of Copenhagen, Copenhagen, Denmark
[3] Department of Neuroanesthesiology, Rigshospitalet, Copenhagen University Hospital,
Copenhagen, Denmark
[4] Department of Neurology, Charité—Universitätsmedizin Berlin, Freie Universität Berlin,
Humboldt-Universität zu Berlin, Berlin Institute of Health, Berlin, Germany
[5] Center for Stroke Research Berlin, Charité—Universitätsmedizin Berlin, Freie Universität Berlin,
Humboldt-Universität zu Berlin, Berlin Institute of Health, Berlin, Germany
[6] Department of Experimental Neurology, Charité—Universitätsmedizin Berlin, Freie Universität
Berlin, Humboldt-Universität zu Berlin, Berlin Institute of Health, Berlin, Germany
[7] Bernstein Center for Computational Neuroscience Berlin, Berlin, Germany
[8] Einstein Center for Neurosciences Berlin, Berlin, Germany

## ABSTRACT

**Background:** Near-death experiences (NDE) occur with imminent death and in situations of stress and danger but are poorly understood. Evidence suggests that NDE are associated with rapid eye movement (REM) sleep intrusion, a feature of narcolepsy. Previous studies further found REM abnormalities and an increased frequency of dream-enacting behavior in migraine patients, as well as an association between migraine with aura and narcolepsy. We therefore investigated if NDE are more common in people with migraine aura.

**Methods:** We recruited 1,037 laypeople from 35 countries and five continents, without any filters except for English language and age ≥18 years, via a crowdsourcing platform. Reports were validated using the Greyson NDE Scale.

**Results:** Eighty-one of 1,037 participants had NDE (7.8%; CI [6.3–9.7%]). There were no significant associations between NDE and age ($p > 0.6$, $t$-test independent samples) or gender ($p > 0.9$, Chi-square test). The only significant association was between NDE and migraine aura: 48 (6.1%) of 783 subjects without migraine aura and 33 (13.0%) of 254 subjects with migraine aura had NDE ($p < 0.001$, odds ratio (OR) = 2.29). In multiple logistic regression analysis, migraine aura remained significant after adjustment for age ($p < 0.001$, OR = 2.31), gender ($p < 0.001$, OR = 2.33), or both ($p < 0.001$, OR = 2.33).

**Conclusions:** In our sample, migraine aura was a predictor of NDE. This indirectly supports the association between NDE and REM intrusion and might have implications for the understanding of NDE, because a variant of spreading depolarization (SD), terminal SD, occurs in humans at the end of life, while a short-lasting variant of SD is considered the pathophysiological correlate of migraine aura.

Corresponding author
Daniel Kondziella,
daniel_kondziella@yahoo.com

# INTRODUCTION

Near-death experiences (NDE) include emotional, self-related, spiritual and mystical perceptions and feelings, occurring in situations close to death or in other situations of imminent physical or emotional danger (*Greyson, 1983*; *Parnia et al., 2014*). Common themes of NDE comprise, but are not restricted to, out-of-body experiences, visual and auditory hallucinations, distortion of time perception, and increased speed of thoughts (*Greyson, 1983*). See *Peinkhofer, Dreier & Kondziella (2019)* for a recent overview on NDE.

The neuronal mechanisms of NDE are poorly understood. Nelson and colleagues previously proposed the concept that rapid eye movement (REM) sleep intrusion and REM related out-of-body experiences could occur at the time of a life-threatening event and might explain many elements of NDE (*Nelson et al., 2006*; *Nelson, Mattingly & Schmitt, 2007*). REM sleep is defined by rapid and random saccadic eye movements, loss of muscle tone, vivid dreaming, and cortical activation as revealed by desynchronization of the scalp electroencephalography (EEG). REM state features can intrude into wakefulness, both in healthy individuals and patients with narcolepsy. This may cause visual and auditory hallucinations at sleep onset (hypnagogic) or upon awakening (hypnopompic) and muscle atonia with sleep paralysis and cataplexy (*Scammell, 2015*). According to the hypothesis of Nelson and colleagues, danger provokes the arousal of neural pathways that, when stimulated, are known to generate REM-associated responses. This was interpreted as a "diathesis-stress model" (*Nelson et al., 2006*; *Long & Holden, 2007*). In this model, an unusually sensitive arousal system (i.e., the diathesis), as evidenced by the experience of REM intrusion, would predispose people to NDE in situations of stress and danger. To test their hypothesis, *Nelson et al. (2006)* conducted a survey comparing a group of individuals with self-reported NDE and an age- and sex-matched control group. The results suggested that episodes of REM intrusion are more common in individuals with NDE.

The study by Nelson et al. has been criticized (*Long & Holden, 2007*), however, which recently inspired us to carry out a follow-up study in a different setting to address some of the criticism (*Kondziella, Dreier & Olsen, 2019*). For example, *Long & Holden (2007)* pointed out that 40% of the people with NDE in the Nelson study denied ever having experienced an episode of REM intrusion, suggesting that there may be a link between the two phenomena, but not a 1:1 relationship. In our crowdsourced survey, 106 of 1,034 participants reported NDE according to a Greyson NDE scale (GNDES) score ≥7, and 50 (47%) of these individuals fulfilled the criteria of REM intrusion according to almost the identical questionnaire that Nelson and colleagues had used (*Kondziella, Dreier & Olsen, 2019*). In contrast, only 17% of individuals without NDE reported REM intrusions. Based on multivariate regression analysis, we found that REM intrusion is a predictor of NDE (*Kondziella, Dreier & Olsen, 2019*). Thus, we

confirmed the results of Nelson and colleagues, but also the limitation that this is not a 1:1 relationship.

A more central point of criticism was related to the control group in Nelson and colleagues' study which consisted mainly of medical personnel, a likely selection bias (*Long & Holden, 2007*). We countered this in our survey with a crowdsourced approach in which the control group originated from the same population as the NDE group (i.e., unprimed laypeople) (*Kondziella, Dreier & Olsen, 2019*). Our survey was announced under the headline "Survey on NDE and (related experiences)," but we did not provide further information about the content of the study. Participants were informed that their monetary reward was fixed, regardless of whether they would report having had an NDE or not. Then, we asked the participants to complete a questionnaire comprising demographic information, followed by the questions about REM intrusion. Subsequently, participants were asked if they ever had experienced an NDE. If not, the survey ended there; if yes, participants were asked in detail about this experience and information about all 16 GNDES items was collected (*Kondziella, Dreier & Olsen, 2019*). In this way, we think that we were able to dispel the previous criticism regarding the control group.

*Long & Holden (2007)* also explained how the questionnaire for REM intrusion could be misinterpreted by people with NDE, possibly leading to an overestimation of the association between REM intrusion and NDE. It is indeed difficult to address this problem with a questionnaire containing only closed questions. Therefore, we also gave our participants the opportunity to describe their experiences in their own words (*Kondziella, Dreier & Olsen, 2019*).

Another approach to address this problem is to investigate if comorbidities of REM intrusion, which might be easier to detect with a questionnaire, are associated with NDE too. In this context, it is interesting that REM sleep abnormalities have been linked to migraine. Thus, recurrent vivid dreams are associated with migraine attacks (*Lippman, 1954*); migraine attacks often occur during REM sleep (*Levitan, 1984*); hallucinations are not infrequent in people with migraine (*Lippman, 1951*, *1953*; *Daniel & Donnet, 2011*); migraine patients exhibit increased REM sleep and prolonged REM sleep latencies (*Drake et al., 1990*); and they show a significantly increased frequency of dream-enacting behavior (*Suzuki et al., 2013*). In addition, several studies found an association between migraine and narcolepsy, a disorder involving REM intrusion (*Dahmen et al., 1999*, *2003*; *Longstreth et al., 2007*; *Suzuki et al., 2015*; *Yang et al., 2017*). For example, *Yang et al. (2017)* found a consistently higher risk of developing narcolepsy in children with migraine compared to those without, and this risk was particularly high in children with migraine with aura.

On this basis, we hypothesized that, analogous to an unusually sensitive arousal system underlying REM intrusion, an increased susceptibility of the brain to spreading depolarization (SD), the assumed pathophysiological correlate of migraine aura (Fig. 1A), could predispose people to NDE. To test this hypothesis, we recruited a large global sample of laypersons and investigated if the lifetime occurrence of migraine aura is more common in people with NDE.

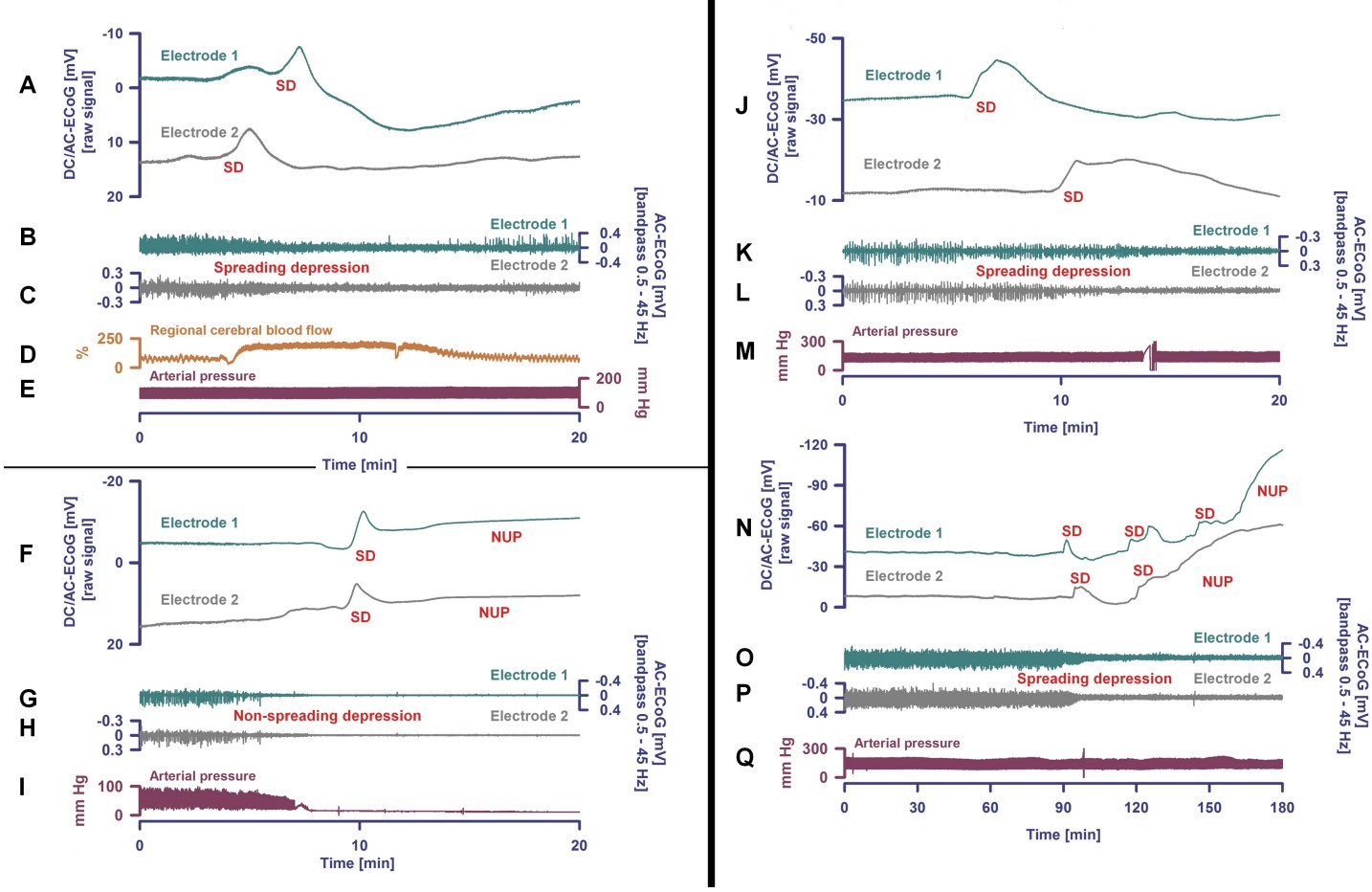

**Figure 1 Spreading depolarization occurs both in migraine aura and in the dying human brain.** Examples from three patients with spreading depolarizations. Patient 1 (A–E). Spreading depolarization (SD) is observed as a large negative direct current (DC) shift propagating between different electrodes (A) (subdural full-band DC/alternate current (AC)-electrocorticography (ECoG) between 0 and 45 Hz, electrode separation: one cm) (*Dreier et al., 2017*). This short-lasting SD was recorded in a patient with aneurysmal subarachnoid hemorrhage (aSAH) in a metabolically largely intact and sufficiently perfused neocortex region. Based on measurements of regional cerebral blood flow (rCBF) using intracarotid [133]Xe and positron emission tomography, blood-oxygen-level dependent (BOLD) imaging with functional magnetic resonance imaging (MRI) and magnetoencephalography (MEG), it is assumed that the SD underlying a migraine aura should be largely similar (*Dreier & Reiffurth, 2015*). The patient's perception of a migraine aura is presumably triggered by the SD-induced spreading depression of spontaneous activity (*Dreier & Reiffurth, 2015*), which is shown here in (B) and (C) as a transient reduction in amplitudes propagating between electrodes (frequency band: 0.5–45 Hz). It should be noted, however, that a patient can only perceive a migraine aura if this spreading depression propagates through an eloquent region of the brain (*Dreier & Reiffurth, 2015*). SD is characterized by the almost complete collapse of ion gradients across cell membranes, causing water influx and an almost complete loss of Gibbs' free energy contained in the ion gradients (*Dreier et al., 2013*). Recovery from SD requires activation of adenosine triphosphate (ATP)-dependent membrane pumps, in particular Na, K-ATPases. Therefore, tissue ATP declines by ca. 50% during SD not only in energy-deprived but also in well-nourished tissue (*Dreier & Reiffurth, 2015*). Consequently, rCBF significantly increases (D) in normal tissue to meet the enhanced energy demand and to clear the tissue of metabolites (measurement of rCBF using an optoelectrode and laser-Doppler flowmetry). The regional hyperemia is variably followed by a mild rCBF decrease (oligemia) during which the vascular reactivity is disturbed. The short initial hypoperfusion is an abnormality here that indicates mild impairment of the neurovascular coupling in the context of aSAH (*Dreier & Reiffurth, 2015*). The arterial blood pressure (E) measured in the radial artery was stable during the SD. Patient 2 (F–I). The second patient died from hepatorenal failure several days after aSAH (*Dreier et al., 2018*). The circulatory arrest is evidenced by the drop in arterial blood pressure (I). About 35 seconds after the circulatory arrest, the AC-ECoG in traces (G) and (H) begin to show the non-spreading depression of spontaneous activity. Phase 2 lasts 95 s at electrode 2 (H). Thereafter, the terminal SD occurs and spreads further from electrode 2 to electrode 1 (electrode separation: one cm) (F). Terminal SD consists of the initial SD component and the late negative ultraslow potential (NUP). It remains speculative if NDE can occur in ECoG phases 1, 2 or 3. According to current knowledge, however, the occurrence of NDE in phases 2 or 3 cannot be ruled out. As explained in the main text, ECoG and scalp electroencephalography (EEG) show a flat line in phase 2, but experiments in animals and brain slices with sophisticated electrophysiological techniques including patch-clamping have shown that the synaptic terminals remain highly active in this phase and that the

**Figure 1** (continued)

neurons are polarized (*Müller & Somjen, 2000*; *Fleidervish et al., 2001*; *Allen, Rossi & Attwell, 2004*; *Revah et al., 2016*). Therefore, we cannot exclude with certainty that patients may experience a perception at this stage. The terminal depolarization takes place in phase 3. It cannot be excluded either that this may be associated with bright light phenomena or tunnel vision similar to what occurs during a migraine aura. Brain cells die only gradually in phase 4 which is characterized by the NUP (F). Patient 3 (J–Q). After onset of the terminal cluster of SDs shown in (J) and (N), this patient with aSAH was found to have lost brainstem reflexes with fixed dilated pupils, indicating the development of brain death (*Dreier et al., 2019*). The cluster starts here at electrode 1 and propagates to electrode 2 (J and N). The first SD occurs in electrically active tissue and therefore causes spreading depression of the spontaneous ECoG activity (K and L). In contrast to Patient 1 (B and C), however, activity depression then persists (O and P). After the first SD, a second SD occurs, which transforms into a NUP (N). In contrast to Patient 2 (F), further SDs are superimposed on the NUP, but their amplitudes become smaller and smaller (N). Like in patient 1 (E), the arterial blood pressure (Q) remains stable during the SDs and the NUP. The patient was terminally extubated 20 h later and shortly thereafter a circulatory arrest developed without further SD (*Dreier et al., 2019*). Data from Patient 2 (*Dreier et al., 2018*) and Patient 3 (*Dreier et al., 2019*) are presented here in abbreviated form to illustrate the pivotal aspects of brain death at the tissue level. The patients were enrolled at the Charité—Universitätsmedizin Berlin in research protocols of invasive neuromonitoring approved by the local ethics committee and written informed consent was obtained from the patients' legally authorized representative, as described previously (*Dreier et al., 2018*, *2019*).

## MATERIALS AND METHODS

### Study design

Our objective was to investigate whether people with a history of migraine aura are more likely to have NDE, and vice versa, than people without migraine aura. We used an online platform, Prolific Academic (https://prolific.ac/), to recruit an international sample of laypeople. Like Amazon's Mechanical Turk, Prolific Academic is a crowdsourcing online platform to recruit human subjects that can be used for research purposes (*Kondziella, Dreier & Olsen, 2019*; *Kondziella, Cheung & Dutta, 2019*) and that compares favorably in terms of data quality, including honesty and diversity of participants (*Peer et al., 2017*). Participants were recruited without any filters except for English language and age ≥18 years, and we excluded participants who had been enrolled in our previous study on NDE and REM intrusion (*Kondziella, Dreier & Olsen, 2019*). The study was announced under the headline "Survey on NDE and headache" using the following text: "We wish to explore the frequency with which NDE occur in the public. This should take no more than 1.5 min on average (a little bit longer, if you have had such an experience, and a little bit less, if you haven't). You will be paid 0.20$ after completing the survey. Please note that we might use your anonymous answers when writing a paper."

From all participants, we collected information about age, gender, place of residence and employment status (data provided automatically by Prolific Academic); if they had frequent headaches; if yes, if these headaches could last longer than 4 h and were associated with visual or non-visual aura (*Kaiser et al., 2019*); if participants ever had an NDE; if yes, if this experience occurred in a truly life-threatening situation or in a situation that just felt so; if the experience was neutral, pleasant or unpleasant; and all participants with an NDE were asked to provide information about all 16 items of the GNDES, the most widely used standardized tool to identify, confirm and characterize NDE in research (*Greyson, 1983*). Like in our previous study (*Kondziella, Dreier & Olsen, 2019*), NDE was defined by a GNDES score ≥7. We evaluated all reports of NDE, irrespective of whether they occurred in truly life-threatening situations or in situations that just felt so. Participants with an

**Table 1 Questionnaire on headaches, migraine aura and near-death experiences.**

Questions about headache (adapted from *Kaiser et al. (2019)*)

- Do you get headaches that are NOT caused by a head injury, hangover, or an illness such as the cold or the flu?
- Do your headaches ever last more than 4 h?
- Have you ever had visual disturbances around the time of your headache? For example, have you ever seen any spots, stars, lines, flashing lights, zigzag lines, or heat waves?
- Around the time of your headaches, have you ever had: Numbness or tingling of your body or face, weakness of your arm leg, face, or half of your body, difficulty speaking, or none of the above.

Questions about near-death experiences

- Near-death experiences can be defined as any conscious perceptual experience, including emotional, self-related, spiritual and/or mystical experiences, occurring in a person close to death or in situations of intense physical or emotional danger. In plain language—near-death experiences are exceptional experiences that you may have when you are dying or feel as if you were dying. Have you ever had such a near-death experience—either during a true life-threatening event or an event that just felt so?
- Was your near-death experience associated with a true life-threatening event or an event that was not life-threatening but felt so?
- If you wish, please describe your experience (this is optional). We are interested to know what you felt, how your experience unfolded over time and in which situation you had your near-death experience.

GNDES (zero to two points for each answer; based on *Greyson (1983)*.

1. Did time seem to speed up or slow down?
2. Were your thoughts speeded up?
3. Did scenes from your past come back to you?
4. Did you suddenly seem to understand everything?
5. Did you have a feeling of peace or pleasantness?*
6. Did you have a feeling of joy?
7. Did you feel a sense of harmony or unity with the universe?
8. Did you see, or feel surrounded by, a brilliant light?
9. Were your senses more vivid than usual?
10. Did you seem to be aware of things going on elsewhere, as if by extrasensory perception or telepathy?
11. Did scenes from the future come to you?
12. Did you feel separated from your body?
13. Did you seem to enter some other, unearthly world?
14. Did you seem to encounter a mystical being or presence or hear an unidentifiable voice?
15. Did you see deceased or religious spirits?
16. Did you come to a border or point of no return?

**Note:**
* In contrast to the Greyson Near-Death Experience Scale (GNDES), we also questioned about unpleasant experiences.

NDE (and those who claimed an NDE but scored six or less points on the GNDES) were also given the opportunity to describe this in their own words (optional). We did not inquire about REM sleep intrusion, which we assessed in our previous study (*Kondziella, Dreier & Olsen, 2019*). Table 1 provides details.

## Statistics

Using a very high population size (300,000,000), a confidence level of 95% and a margin of error of 5%, we estimated the required sample size to be 384 participants. However, since previous studies have estimated the frequency with which NDE occur in the public to
be 5–10%, including our own on NDE and REM intrusion (*Kondziella, Dreier & Olsen, 2019*), we decided to enroll approximately 1,000 participants to identify an estimated number of 100 individuals with an NDE.

In univariate analysis, associations between potential predictors (age, gender, migraine aura) for NDE were examined using Chi-square test and *t*-test for independent samples. Additionally, we used multivariable logistic regression to analyze the association between migraine aura and NDE adjusted for age and gender. The level of significance was 0.05 (two-sided) for all statistical tests. Statistical analysis was performed with SPSS 23.0 (IBM, Armonk, NY, USA).

### Ethics

Participants gave consent for publication of their anonymous data. Participation was voluntary, anonymous and restricted to those aged 18 years or older. Participants received a monetary reimbursement after completing the survey, in accordance with the Prolific Academic's *ethical rewards* principle (≥$6.50/h). The Ethics Committee of the Capital Region of Denmark waives approval for online surveys (Section 14 (1) of the Committee Act. 2; http://www.nvk.dk/english).

### Data availability statement

The de-identified raw data are provided in the Supplemental Files.

## RESULTS

We recruited 1,037 laypeople from 35 countries and five continents (mean age: 31 years, standard deviation: 11.1 years, median age: 28 years, interquartile range (IQR): 23–36 years; 76% fully or part-time employed or in training), most of which were residing in Europe and North America (Fig. S1). Six participants identified themselves as transgender, 531 (52%) as female, and 500 (48%) as male.

### Near-death experiences: frequency and phenomenology

A total of 286 participants (28% (95% CI [25–30%])) claimed an NDE. The most frequent symptoms were abnormal time perception (faster or slower than normal; reported by 257 participants; 90%); extraordinary speed of thoughts ($n = 169$; 59%); exceptional vivid senses ($n = 165$; 58%); and feeling separated from one's body, including out-of-body experiences ($n = 113$; 40%). Participants perceived the situation in which they made their experience slightly more often as truly life-threatening ($n = 165$; 58%) than not ($n = 121$; 42%).

However, only 81 of 286 individuals who claimed an NDE reached the threshold of ≥7 points on the GNDES (28% (95% CI [23–34%])). Hence, confirmed NDE were reported by 81 of 1,037 participants (8% (95% CI [6.3–9.7%])) (Figs. 2 and 3). Confirmed NDE were perceived much more often as pleasant ($n = 29$; 49%) than experiences that did not qualify as NDE according to the GNDES ($n = 21$; 13%; $p < 0.0001$; Chi-square test; neutral experiences excluded). Table 2 provides selected written reports from participants with an NDE of ≥7 GNDES points and Table 3 from participants with <7 GNDES points.

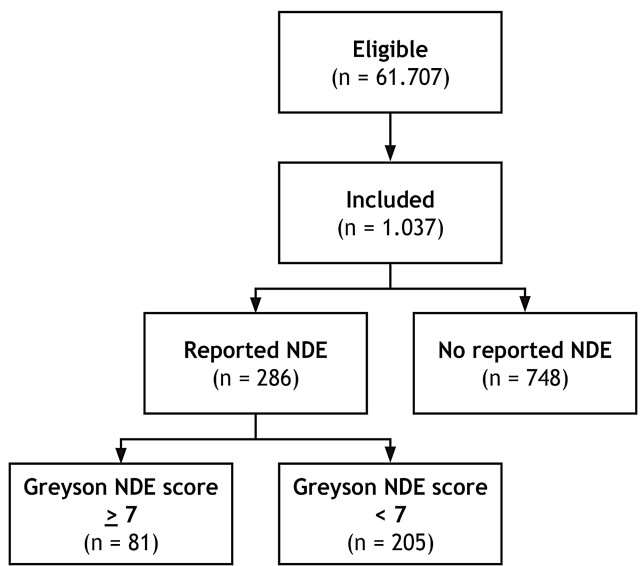

**Figure 2 Schematic overview of the study design.** Of 61.707 eligible lay people registered with Prolific Academic (https://prolific.ac/; accessed on February 4, 2019), we enrolled 1,037 participants; 81 (7.8% (95% CI [6.3–9.7%])) of whom reported a near-death experience that fulfilled established criteria (Greyson Near-Death Experience Scale score of 7 or higher). *n*, number of participants; NDE, near-death experience.

## Headache and migraine aura

Seven-hundred-twenty of 1,037 individuals (69%) answered "yes" to the following question about a primary headache disorder: "Do you get headaches that are NOT caused by a head injury, hangover, or an illness such as the cold or the flu?" The male-to-female ratio of people who responded "yes" to this question was 1:1.3. Two-hundred-fifty of 1,037 individuals (24%) fulfilled criteria (*Kaiser et al., 2019*) of having experienced a migraine aura at any point during their lifetime. Individuals could have different types of migraine aura. Two-hundred-thirty of 254 (91%) individuals reported having had visual auras, 60 (24%) somato-sensory auras, 49 (19%) motor auras and 21 (8%) aphasic/dysarthric auras. Hundred-seventy-four of 531 women (33%) had a migraine aura and 77 of 500 men (15%). This difference was statistically significant ($p < 0.001$, Chi-square test; male-to-female ratio: 1:2.2). People with migraine aura were slightly older than people without migraine aura (median age: 30 (IQR: 24–38) years vs median age: 28 (IQR: 22–36) years, $p = 0.005$, Mann–Whitney Rank Sum Test).

## Near-death experiences and evidence of migraine aura

There were no significant associations between confirmed NDE and age ($p > 0.6$, *t*-test independent samples) or gender ($p > 0.9$, Chi-square test). The only significant association was between confirmed NDE and migraine aura: 48 (6.1%) of 783 subjects without migraine aura and 33 (13.0%) of 254 subjects with migraine aura had experienced an NDE ($p < 0.001$, odds ratio (OR) = 2.29). In multiple logistic regression analysis with age, gender and the interaction of age and gender, none of these potential predictors was

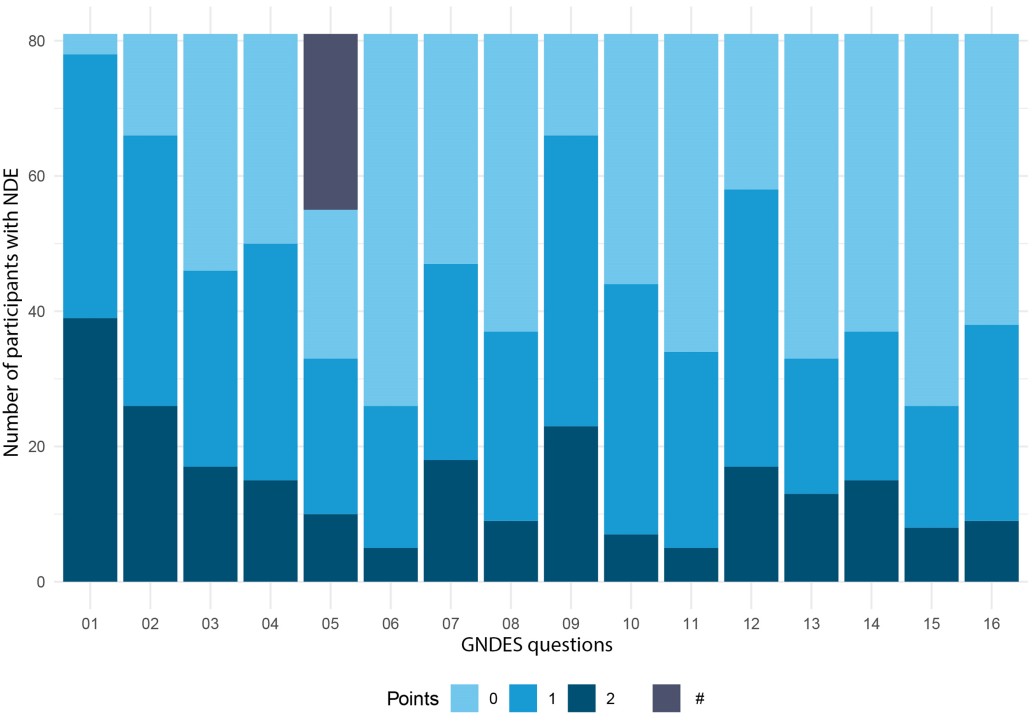

**Figure 3 Experiences of participants with NDE.** This figure illustrates the experiences of 81 participants with NDE confirmed by the Greyson Near-Death Experience Scale (GNDES), that is, those with seven points or more on the GNDES. See Table 1 for GNDES questions 1–16. Each question is given zero to two points (0 = "no"; 1 and 2 = "yes," weighted according to the intensity of the feature experienced). #—unpleasant experience (question 5); in contrast to the GNDES, we also inquired about unpleasant experiences.

significant. However, migraine aura remained significant after adjustment for age ($p < 0.001$, OR = 2.31), gender ($p < 0.001$, OR = 2.33), and both age and gender ($p < 0.001$, OR = 2.33).

# DISCUSSION

## The prevalence of NDE

The prevalence of individuals with an NDE is estimated at about 4–8% in the general population (*Gallup & Proctor, 1982*; *Knoblauch, Schmied & Schnettler, 2001*; *Perera, Padmasekara & Belanti, 2005*; *Facco & Agrillo, 2012*; *Chandradasa et al., 2018*). In our survey, it was 8%. We found a prevalence of 10% using the same criteria in our previous crowdsourcing online survey on NDE and REM intrusion (*Kondziella, Dreier & Olsen, 2019*), indicating that this prevalence is quite robust. Unlike most previous reports in which NDE were almost always associated with peace and well-being (*Thonnard et al., 2013*; *Charland-Verville et al., 2014*; *Martial et al., 2017, 2018*; *Cassol et al., 2018*), we confirmed recent findings that many people find their NDE unpleasant (*Charland-Verville et al., 2015*; *Kondziella, Dreier & Olsen, 2019*; *Cassol et al., 2019*). However, experiences with the cut-off score of ≥7 GNDES points were reported more often as pleasant (49%) than experiences with a lower score (13%).

**Table 2 Selected reports from participants with an experience that reached the threshold of ≥7 points on the Greyson Near-Death Experience Scale (GNDES) and qualified as an NDE.** The last two comments describe experiences during ingestion of ketamine (which has been suggested as the chemical most likely to cause drug-induced near-death experiences (*Martial et al., 2019*)) and REM sleep disturbance (which has been identified in another recent study as a likely mechanism of near-death experiences (*Kondziella, Dreier & Olsen, 2019*)). Comments are edited for clarity and spelling.

- (After a suicide attempt) I spend 7 days [in the intensive care unit]. I felt that I did no longer exist in my body; everything went fast as hell and I saw my life (passing before me). I felt that I was moving but it wasn't like any movement that I had known before. I found myself in the light, a very bright place, the whitest white mixed with energy, as if it was almost alive. I saw three luminous figures coming toward me. I was talking with them without using words; it was much easier and more efficient like talking with feelings and unused senses. All what happened was very personal; I (was being shown images and) received an enormous amount of information, but I am unable to explain it, as if all was spoken to me in a non-existent language that (nobody understands) "here" but everyone once "there." You just KNOW things about yourself, the nature of the world and people. It was beautiful, full of love, and it was so simple. One of the individuals was younger than the others and loved to laugh; it was as if he wanted to say that my suicide attempt wasn't so serious any longer, nor was anything else serious. It also felt like they knew that I wanted to be back—not on earth but somewhere else. *38 years, female, migraine auras (visual, aphasic); NDE 23; life-threatening*

- All pain, fears, worries and suffering disappeared. It was an incredibly pleasant feeling, warm and light. I felt unbelievable peace and wanted to remain there, but I was told my moment had not arrived yet and I had to return. When I returned, I felt very secure and knew that death was not to be feared. I remember it as if it was yesterday. *44 years, female; headache without aura; NDE 17; not life-threatening*

- My vision became spherical and I could see and understand everything at once. I also felt as if I was speeding towards a light. I knew if I went to the light I wouldn't come back. It took an enormous effort to change direction and get back into my body. *50 years, female; migraine auras (visual, aphasic); NDE 15; not life-threatening*

- I was bleeding heavily and began to lose all sense of my physical body. There was this incredible sense of peace and harmony, as if all trouble and stress was gone, and if I just let go, it would never return. I then forced myself to come back to consciousness because I knew I was needed at home. I don't think it lasted more than a minute, but it felt much longer. *56 years, female; migraine aura (visual); NDE 8; life-threatening*

- I was almost drowning, when I heard voices in my head telling me how to save myself. My life flashed before my eyes and I saw myself simultaneously being above water level. *25 years, male; migraine aura (visual); NDE 13; life-threatening*

- It did feel very real. Like I was hyperaware of everything around me. Things had a glow. Not just the beings but everything had this muted glow. *48 years, female; migraine auras (visual, aphasic); NDE 9; life-threatening*

- (During an anaphylactic shock) I saw a brighter and brighter light. I fell on the ground but barely felt my body falling. I couldn't feel anything about my body. The room was still the same but there was a light or white smoke. I saw a beautiful person with blonde, curly and shoulder-long hair. Her face had thin traits. The eyes were looking more like smoke than actual eyes. It was a person made of light and colors (like a sort of rainbow or these angels in video clips), protective, smooth, nice and peaceful. The person couldn't talk but gave me her hand and I started to go with her. Then I felt horrible pain. I briefly felt floating over my body. Then I woke up. I think a saw an angel who was in charge to check if it was my time to go, and if so, to lead me to another place. Maybe it was my deceased 40 years ago grandmother, but I'm not sure. *27 years, female; no headaches; NDE 21; life-threatening*

- I was sleeping, and something woke me up. I felt someone watching me, but I was alone. I couldn't breathe. I tried fighting it and felt weird, like I was outside of my body. Time slowed down. Suddenly, it all disappeared. It lasted maybe a minute, but I felt like it was hours. *28-years, female, headache without aura; NDE 8; not life-threatening*

- My experience was induced by Ketamine at a rave party. *33 years, male, headache without aura; NDE 10; not life-threatening*

**Table 3 Selected reports from participants with an experience below the threshold of ≥7 points on Greyson Near-Death Experience Scale (GNDES) and that did not qualify as an NDE.** Comments are edited for clarity and spelling.

- It was an ordinary circumstance that turned serious quickly. My throat closed and would not open, no matter how much I tried. It closed so long that I started to black out. At first, I was fearful. Then, I remember feeling a great sense of peace and acceptance of death. *33 years, male; headache without aura; NDE 4; life-threatening*
- I was in a playground accident aged 12 and drifted in and out of consciousness. Everything seemed to happen incredibly quickly, and I was unable to distinguish between the real emergency respondents and members of family, both deceased and present. I could not feel contact with the ground and believed that I was floating above it. *32 years, male; headache without aura; NDE 6; life-threatening*
- I got in the water and suddenly there wasn't anything under my feet, and I was drowning. I saw my life flash quickly before me. It felt very fast but at the same time also very slow. *22 years, female; migraine aura (sensory); NDE 5; life-threatening*
- I nearly drowned, and I became incredibly comfortable and at peace with myself. *53 years, male, headache without aura; NDE 6; life-threatening*

## Migraine aura is a predictor of NDE

Migraine aura was a predictor of NDE in our sample. This association was very stable. Regardless of whether either no adjustment, an adjustment for age, for sex or for both was performed, the ORs for migraine aura only varied between 2.29 and 2.33. However, a potential limitation of our study is the announcement of the internet query in which we stated that we would investigate for NDE and headache. This might have attracted more people with NDE and headache. The overall prevalence for all types of primary headache, including tension-type headache, was 69% in our survey. Tension-type headache is the most common form of headache (*Jensen, 2018*). Its aggregate prevalence in the general population across different studies was 38% (*Jensen, 2018*). Yet, in a population-based study in Denmark, a much higher lifetime prevalence of 78% was found (*Lyngberg et al., 2005*; *Jensen, 2018*). The high prevalence of primary headaches in our survey is hence within the realm of possibility but raises the question if we have attracted a disproportionate number of people with headache. This could include people with migraine with aura. The observation that 24% of the participants in our survey met criteria for a migraine aura, while population-based studies have estimated this prevalence at only 4% in the general population, renders this indeed likely (*Russel et al., 1995*). The young average age, typical of an Internet-based study, could have contributed to over-representation of migraineurs with aura. The way we phrased our headache questions could be another reason, as we did not intend to validate a migraine diagnosis according to established criteria (*Kaiser et al., 2019*). Instead, we used a more inclusive approach to identify people with a high likelihood of having migraine aura because we were not interested in migraine per se but rather in migraine aura as a possible predictor for an NDE (*Kaiser et al., 2019*). Since population-based studies suggest that spontaneous migraine aura is four times less common in people without typical migraine headache than in people with typical migraine headache (*Russel et al., 1995*), it is unlikely that the over-representation of people with migraine aura in our survey resulted from the fact that we also included people with migraine aura without typical migraine headache. However, we did not ask whether the aura symptoms lasted at least 5 min (It should be noted that the threshold of >5 min to classify as migraine aura is arbitrary). Accordingly, in

humans it has been shown that SD, the pathophysiological correlate of migraine aura, may occur in spatially very limited fields and that the propagation speed in the cortical tissue ranges between ~2 and nine mm/min (*Woitzik et al., 2013*). On one hand, this could have contributed to the discrepancy between our data and population-based migraine studies. On the other hand, the male-to-female ratio in individuals with migraine aura was 1:2.2 in our survey, which is well in line with the results of population-based studies and supports that we indeed detected variants of migraine aura (*Russel et al., 1995*). In contrast, the male-to-female ratio of a primary headache disorder, be it tension-type headache, migraine or a rarer headache, was 1:1.3 overall. This ratio is again well in line with the assumption that the vast majority of primary headache sufferers in our survey had episodic tension-type headache (*Jensen, 2018*).

The recurrent burden of headache may have increased motivation to participate in our survey, although this remains entirely speculative. The important question, however, is whether the combination of NDE and migraine aura disproportionately increased the motivation of affected people to join our study. Mathematically, we deal with three random factors: migraine aura (yes/no), NDE (yes/no), and participation (yes/no). The twofold dependencies between participation and migraine aura or NDE appear unproblematic. In contrast, a threefold dependency between participation, migraine aura and NDE could have produced a spurious association. However, we consider this unlikely because, for instance, the entire survey was finished during such a short time frame (i.e., within 3 h after posting the survey online) that word-of-mouth communication of the survey's topic seems very unlikely.

Internet-based surveys and more traditional mail-based questionnaires or laboratory-based studies each have their advantages and disadvantages (*Kaiser et al., 2019*). We suggest that a combination of the different approaches is more meaningful than using just one method (*Kondziella, Dreier & Olsen, 2019*). On one side, complex clinical and ethical concepts cannot be fully captured by an online survey (*Woods et al., 2015*; *Peer et al., 2017*). For instance, we did not inquire about precipitating factors/contexts in which participants had experienced their NDE (although we did so in our previous study on NDE and REM intrusion (*Kondziella, Dreier & Olsen, 2019*)). Future non-internet-based studies will therefore be necessary to verify that NDE and migraine aura are indeed associated. On the other side, the anonymous character of a crowdsourcing online survey decreases the influence of psychological bias (*Woods et al., 2015*; *Peer et al., 2017*), because there is no incentive to satisfy the investigator by exaggerating or inventing memories. There was no monetary incentive in our survey either, since we instructed participants that their reimbursement would be the same regardless of whether they reported an NDE or headache or not. In addition, we recruited a much larger sample than would have been feasible during a conventional survey. Although participants from Europe and North America made up the largest share, ours was indeed a global sample with people from 35 countries and five continents.

## NDE and the neurobiology of dying

In the largest prospective multi-center observational trial on AWAreness during Resuscitation (AWARE), 46% of 140 survivors reported memories following their

cardiac arrest with seven major cognitive themes (*Parnia et al., 2014*). Nine percent of the survivors met the criteria for an NDE according to the GNDES. Two percent described awareness with explicit memories of "seeing" or "hearing" real events related to their resuscitation. Importantly, one patient had a verifiable period of conscious awareness during which time cerebral function was not expected (*Parnia et al., 2014*). Although speculative, it seems likely that there is a neurobiological basis for this observation (*Nelson et al., 2006*; *Parnia et al., 2014*; *Martial et al., 2019*; *Peinkhofer, Dreier & Kondziella, 2019*). The pathophysiological events that occur during the process of dying are of obvious interest in this regard (*Vrselja et al., 2019*). The transition from life to death is thus characterized by four major events: loss of circulation, loss of respiration, loss of spontaneous electrocorticography (ECoG) activity, and a terminal SD without repolarization. These four events occur always, but not necessarily in the same order (*Dreier et al., 2018*, *2019*; *Carlson et al., 2018*). In the most common scenario, arrest of systemic circulation, respiration and ECoG activity develops more or less simultaneously, while terminal SD follows the complete arrest of ECoG activity with a latency of 13–266 s (*Dreier et al., 2018*). Along this sequence, the invasively recorded direct current (DC)/alternate (AC)-ECoG activity can be roughly divided into four different phases which are illustrated with an original recording from a previous study (*Dreier et al., 2018*) in Fig. 1B: In phase 1, spontaneous ECoG activity is still measurable; phase 2 is characterized by a complete loss of ECoG activity starting simultaneously in different cortical regions and layers, which is referred to as non-spreading depression of spontaneous activity (*Dreier, 2011*); in phase 3, the terminal SD starts but, from a phenomenologically point of view, is initially similar to SD spreading in healthy grey brain matter (Fig. 1A) (*Dreier & Reiffurth, 2015*; *Hartings et al., 2017a*); and finally, in phase 4 a negative ultraslow potential signals the second phase of terminal SD (*Oliveira-Ferreira et al., 2010*; *Hartings et al., 2017b*; *Dreier et al., 2018*, *2019*; *Lückl et al., 2018*; *Carlson et al., 2018*).

The pertinent question arising from the AWARE study is whether phase 2 and (the transition to) phase 3 are compatible with a conscious perception by the patient—and hence, might contribute to the pathophysiological mechanisms of an NDE. On closer examination of the experimental data, it is interesting that the non-spreading depression of spontaneous ECoG activity in phase 2 does not result from a loss of synaptic activity, but on the contrary from vesicular release of various transmitters, including GABA and glutamate, leading to an incoherent, massive increase in miniature excitatory and inhibitory postsynaptic potentials that replace the normal postsynaptic potentials (*Fleidervish et al., 2001*; *Allen, Rossi & Attwell, 2004*; *Revah et al., 2016*). This probably leads to gradual depletion of the releasable pool of vesicles in the synaptic terminals, and thereby significantly distorts neuronal interactions (*Fleidervish et al., 2001*; *Revah et al., 2016*) (Not only are the miniature potentials small, but the abnormal neuronal desynchronization also prevents these potentials from summing-up, which precludes their measurement using comparatively insensitive methods such as subdural and intracortical ECoG or the even cruder scalp EEG). Initially, neurons are hyperpolarized (*Tanaka et al., 1997*; *Müller & Somjen, 2000*). Over time, intracellular calcium and extracellular potassium concentrations gradually increase, while extracellular pH decreases (*Kraig,*

*Ferreira-Filho & Nicholson, 1983*; *Mutch & Hansen, 1984*; *Nedergaard & Hansen, 1993*; *Erdemli, Xu & Krnjevic, 1998*; *Müller & Somjen, 2000*; *Dreier et al., 2002*). Eventually, hyperpolarization turns into neuronal depolarization. When the adenosine triphosphate (ATP) stores are exhausted, ATP-dependent membrane pumps such as the Na, K-ATPase become unable to replenish the leaking ions. Consequently, SD erupts at one or more sites of the cortical tissue and spreads into the environment as a giant wave of depolarization. It is important to understand that this terminal SD marks the onset of the toxic cellular changes that ultimately lead to death, but it is not a marker of death per se, since the SD is reversible—to a certain point—with restoration of the circulation (*Hossmann & Sato, 1970*; *Heiss & Rosner, 1983*; *Memezawa, Smith & Siesjö, 1992*; *Ayad, Verity & Rubinstein, 1994*; *Shen et al., 2005*; *Pignataro, Simon & Boison, 2007*; *Nozari et al., 2010*; *Lückl et al., 2018*). Thus, in contrast to what happens during coma or sedation, when the brain dies, it undergoes a massive and unstoppable depolarization process (and hence, a very last state of "activation") (*Dreier, 2011*).

Returning to the association between NDE and REM intrusion, it would be interesting to know if also a link exists between miniature excitatory/inhibitory postsynaptic potentials and REM sleep. Information is scarce, but there is indeed evidence that these potentials occur in the healthy brain and are involved in the sleep-wake cycle and both REM and non-REM sleep (*Yang & Brown, 2014*; *Christensen et al., 2014*; *Sangare et al., 2016*). Yet, the connection between these potentials in healthy people, on one hand, and disordered neuronal processing, including NDE, on the other hand, has never been properly investigated.

Another unsolved question is if terminal SD could produce bright light phenomena and tunnel vision similar to what happens during a migraine aura, when SD spread through healthy cortical tissue. In this context, it is particularly thought-provoking that terminal SD is not always the final event, but data from so far three patients indicate that terminal SD can sometimes indeed precede circulatory arrest and initiate a spreading depression of spontaneous activity like that in migraineurs with aura (Fig. 1C) (*Dreier et al., 2018*, *2019*; *Carlson et al., 2018*). In contrast to migraine aura, activity then remains depressed at the time of cardiac death.

It is important to bear in mind that virtually all humans (and all animals, including insects (*Spong, Dreier & Robertson, 2017*)) undergo terminal SD at the end of their life, whereas only a minority of people have a migraine aura during their lifetime. Hence, although terminal SD may play a role in the development of NDE, migraine aura during lifetime is probably not required for having an NDE with a bright light at the end of life. However, people with a propensity for migraine aura may be more likely to experience terminal SD while the brain is still electrically active (Fig. 1C). Thus, if terminal SD facilitates NDE, this would suggest that the event of a terminal SD can still be perceived and remembered.

To substantiate or dismiss these speculations, it would be necessary to fully understand how the changing polarization states of approximately 20 billion neurons in the neocortex (*Mortensen et al., 2014*) create the conscious awareness of an individual, an area of intense but unsolved research (*Owen et al., 2006*; *Giacino et al., 2014*; *Kondziella et al.,*

*2016*; *Paulson et al., 2017*; *Demertzi et al., 2019*). This seems important because of the increasing practice of organ donation after cardio-circulatory death (DCD). In countries where DCD is practiced, physicians have reached consensus that death should occur somewhere between a few seconds and 10 min after loss of circulatory function (*Boucek et al., 2008*; *Stiegler et al., 2012*; *Dhanani et al., 2012*; *Van Veen et al., 2018*). Thus, a survey on postmortem organ donation in the framework of the CENTER-TBI study recently revealed that as many as 10 out of 64 centers (16%) in Europe and Israel immediately begin organ retrieval from the donor after a "flat line electrocardiogram" is detected on the monitor (*Van Veen et al., 2018*). Critical voices have been raised, however (*Rady & Verheijde, 2016*; *Youngner & Hyun, 2019*). Due to the above-mentioned uncertainties in our understanding of the dying process, we think it is indeed prudent to consider if organ removal should first be permitted when the neurons in the donor's brain no longer exhibit synaptic transmission and alterations of their polarization state. In other words, organ harvesting should perhaps be postponed until the donor's entire brain has unmistakably reached the negative ultraslow potential phase of terminal SD. It follows that a better understanding of NDE may be relevant to protect the interests of potential organ donors in the context of DCD.

## CONCLUSIONS AND FUTURE DIRECTIONS

In a large global sample of unprimed laypeople, migraine aura was significantly associated with NDE, even after multivariate adjustment. The connection between migraine aura, REM intrusion and NDE is complex. For instance, the brainstem plays an important role in REM intrusion, and dream-like hallucinations such as those in REM sleep are known from people with lesions near the meso-pontine paramedian reticular formation and the midbrain cerebral peduncles (i.e., peduncular hallucinations) (*Galetta & Prasad, 2017*), suggesting that dysfunction of the REM-inhibiting serotonergic dorsal raphe nuclei and the noradrenergic locus coeruleus facilitates REM intrusion (*Hobson, McCarley & Wyzinski, 1975*; *Manford & Andermann, 1998*; *Kayama & Koyama, 2003*; *De Lecea, Carter & Adamantidis, 2012*). A large body of evidence further indicates that the brainstem also plays an important role in the pathogenesis of migraine (*Akerman, Holland & Goadsby, 2011*). Moreover, REM sleep abnormalities have been described in migraineurs; and several reports have substantiated the notion that migraine, in particular migraine with aura, is associated with narcolepsy (*Lippman, 1951*; *Levitan, 1984*; *Drake et al., 1990*; *Dahmen et al., 1999*, *2003*; *Longstreth et al., 2007*; *Suzuki et al., 2013*, *2015*; *Yang et al., 2017*) and hallucinations (*Lippman, 1951*, *1953*; *Daniel & Donnet, 2011*). Hence, although a propensity for REM intrusion is neither necessary, nor sufficient, for having NDE (*Britton & Bootzin, 2004*; *Lopez et al., 2006*; *Long & Holden, 2007*), we and others have suggested that REM intrusion is a predictor of NDE (*Nelson et al., 2006*; *Kondziella, Dreier & Olsen, 2019*). In the present study we found that migraine aura is also a predictor of NDE. The relationship between NDE and migraine aura raises many novel questions which deserve further investigations. In the broadest sense, excitation/inhibition imbalance across different brain structures is likely to play a role (*Van den Maagdenberg et al., 2004*; *Tottene et al., 2009*; *Ambrosini et al., 2016*). However, migraine

aura also has an important vascular component that is particularly interesting for the study of NDE and the dying brain and further increases the complexity of these phenomena and their interactions (*Van den Maagdenberg et al., 2004*; *Tottene et al., 2009*; *Dreier & Reiffurth, 2015*).

### Funding

The authors received funding from RH Forskningspulje; R143-A6132-B3632, Savværksejer Jeppe Juhl og Hustru Ovita Juhls Mindelegat; 27062019, Deutsche Forschungsgemeinschaft (DFG); DFG DR 323/5-1, DFG DR 323/10-1 and FP7; 602150. The funders had no role in study design, data collection and analysis, decision to publish, or preparation of the manuscript.

### Grant Disclosures

The following grant information was disclosed by the authors:
RH Forskningspulje: R143-A6132-B3632.
Savværksejer Jeppe Juhl og Hustru Ovita Juhls Mindelegat: 27062019.
Deutsche Forschungsgemeinschaft: DFG DR 323/5-1, DFG DR 323/10-1 and FP7; 602150.

### Competing Interests

The authors declare that they have no competing interests.

### Author Contributions

- Daniel Kondziella conceived and designed the experiments, performed the experiments, analyzed the data, contributed reagents/materials/analysis tools, prepared figures and/or tables, authored or reviewed drafts of the paper, approved the final draft.
- Markus Harboe Olsen analyzed the data, prepared figures and/or tables, authored or reviewed drafts of the paper, approved the final draft.
- Coline L. Lemale analyzed the data, prepared figures and/or tables, authored or reviewed drafts of the paper, approved the final draft.
- Jens P. Dreier analyzed the data, contributed reagents/materials/analysis tools, authored or reviewed drafts of the paper, approved the final draft.

### Human Ethics

The following information was supplied relating to ethical approvals (i.e., approving body and any reference numbers):

The Ethics Committee of the Capital Region of Denmark waives approval for online surveys (Section 14 (1) of the Committee Act. 2).

### Data Availability

All anonymized raw study data are available in a Supplemental File.

## Supplemental Information

Supplemental information for this article can be found online at http://dx.doi.org/10.7717/peerj.8202#supplemental-information.

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
