# Peer review of "Migraine aura, a predictor of near-death experiences in a crowdsourced study"

_PeerJ, doi:10.7717/peerj.8202_

## Round 0.1 · original submission · Minor Revisions

The authors need to address the criticisms regarding the experimental design and provide more details on the methods, including details on the frequencies of the NDE features and the context in which the participants lived their NDE. The introduction needs improvement and references have to be carefully revised. The discussion should address objections to the hypothesized association between REM intrusion and NDEs as indicated by Reviewer #1.

Reviewer 1 ·

Basic reporting

The writing is clear and unambiguous. The references show sufficient familiarity with the background and context. There are several problems with the reference list, but they are trivial and easily corrected.
The figures are mostly relevant and well labeled,.but Figure 3 ("Map showing places of residency of survey participants") is irrelevant and does not contribute anything to the manuscript and should be deleted.

Experimental design

The experimental design is adequate. There are substantial limitations to the online survey methodology in terms of biased sampling and inability to capture complex concepts, but the authors acknowledge those limitations on lines 261-266 and 302.
Research question is adequately defined and meaningful.
Methods are adequately described.

Validity of the findings

Adequate data are provided for the most part. However, the authors never described their criteria for REM intrusion. They listed in Table 1 the questions on headaches, migraine aura, and NDEs, but they never mentioned the specific questions on REM intrusion, nor did they mention their criteria (i.e., how many of those REM intrusion questions had to be endorsed to fulfill the criteria). Without that information, readers cannot judge how much the REM questions may have overlapped with the NDE questions, leading to a tautological association.
Speculation is not always identified as such. On line 321, the authors write that “one must assume that there has to be a neurological basis” for conscious awareness when cerebral function was not expected. Why must one assume that? This assumption is particularly questionable in light of their citation in the next sentence of the study of restoration of cellular function in a pig brain by Vrselja, et al. (2019), in which the authors explicitly disavowed any implication that this effect had anything to do with consciousness when they wrote: “it is important to distinguish between resuscitation of neurophysiological activity and recovery of integrated brain functions (that is, neurological recovery). The observed restoration of molecular and cellular processes following 4 h of global anoxia or ischaemia should not be extrapolated to signify resurgence of normal brain function. Quite the opposite: at no point did we observe the kind of organized global electrical activity associated with awareness, perception, or other higher-order brain functions.” The present authors, then, should elaborate on the rationale for their assumption. As they acknowledge on lines 392-395, we do not understand how changing polarization states of neocortical neurons create consciousness. And yet, they assume that they do create consciousness somehow.
Additionally, their Discussion should address objections to the hypothesized association between REM intrusion and NDEs, such as the occurrence of NDEs in patients receiving general anesthetics and other drugs that suppress REM activity, and the finding of Britton & Bootzin (2016) of increased REM latency in people who have NDEs.

Additional comments

In addition to the substantive issues mentioned above, the style of references is inconsistent. In some journal article cited, the article title in entirely in lower case, whereas in others, most words are capitalized. Likewise, some journal titles are entirely in lower case, whereas others are capitalized.
On lines 597-598, the authors list a reference to Long & Holden, 2007. On lines 599-600, they list the identical reference, this time citing it as Long & JM, 2007. The second (incorrect) reference should be deleted. On line 107, the authors cited "Long & JM, 2007" but that should be "Long & Holden, 2007".
On line 625, the authors list the Martial, et al., 2019 paper as “In press”. It is no longer in press, but was published in Volume 69.

Reviewer 2 ·

Basic reporting

The paper is well-structured, clear, and well-written, however, I would suggest to the authors to carefully go through their manuscript to correct minor typos and to be consistent along the whole manuscript. For example, they wrote “laypeople” and a bit further “lay people”, line 276 a bracket is missing, etc. Please also double check the references for typos too.

In addition, the numbering of the Figures are not correct (no consistency between the manuscript text and the attached Figures). Please ensure that Tables and Figures are properly referred within the main text.

Experimental design

Kondziella and colleagues conducted an interesting study, as it is the first study to properly assess the potential association between NDEs and migraine. They discuss here a fascinating topic in the science of consciousness, the NDEs, which has not received the scientific and medical attention it deserves. Its strength resides mainly in the number of individuals included, the method used, an interesting discussion and of course its results. I think that this study is important and suitable for this journal.

This is a good paper, albeit some improvements can be made:
- Although it is important to describe the link between REM sleep, migraine aura and NDEs -as they did in the introduction, I think the introduction could be improved by being more concise on this aspect of the introduction. This could leave space for introducing a bit more the literature concerning the presence of hallucinations in migraine patients. In particular, it has been shown that hallucinations are not infrequent in people who have migraine and it would be worthwhile to develop more this literature.
- In the ‘Near-death experiences and evidence of migraine aura’ section of the Materials & Method, it is not clearly stated that the authors also included the experiencers reporting NDEs in a context where they “just” felt that they will die while there was no true life-threatening context. Please more detailed the inclusion criteria somewhere in the Materials & Method.
- Do the authors have more details regarding the context in which the participants lived their NDE? To my opinion, a limitation of this study is that they did not get (self-reported) details regarding the precipitating factors/contexts in which experiencers have experienced the NDE. Indeed, the sample of people who claimed that they “just” felt that they will die while there was no true life-threatening context could actually include a very broad variety of precipitating factors/contexts. Indeed, my own expertise gives me the opportunity to attest that some people might have felt an impression of dying while they actually “only” fainted. According to me, this should clearly be stated in the manuscript.
- Interestingly, the most frequently reported NDE features are ‘abnormal time perception’, ‘extraordinary speed of thoughts’, ‘exceptional vivid senses’, and ‘feeling separated from one’s body’. This frequency pattern might be discussed, as they are not completely consistent with available studies investigating features frequency using standardized scales in NDE reports. Indeed, for example, ‘exceptional vivid senses’ and ‘extraordinary speed of thoughts’ are features reported less often than the feeling of peace/well-being and out-of-body experiences. Moreover, it would be interesting to detail the frequencies of the NDE features reported only by the sample of experiencers who reached 7/32 on the GNDES -and, if the authors wish, maybe have a Figure or Table to illustrate it as well.
- In the first paragraph of their discussion, the authors discussed unpleasant NDEs. That’s interesting and relevant because very little is known on unpleasant NDEs. I would suggest to add the most recent publication studying distressing NDEs, which is Cassol et al. (2019) published in Memory. I think it is very relevant here.

Validity of the findings

The statistics used by the authors are appropriate from what I can tell. The authors' conclusions are well stated and linked to their research question.

Reviewer 3 ·

Basic reporting

By assessing the prevalence of NDEs in people presenting or not migraines with aura, this paper adds an interesting piece to the NDE literature. However, my general feeling about the introduction is that authors focus extensively on how their previous study about REM sleep intrusion addresses the biases of the study by Nelson et al. It would benefit the reader to hear a bit more about the content of NDE and NDE-like memories from previous research.

Experimental design

no comment

Validity of the findings

no comment.

Additional comments

Abstract:
-In the abstract, authors could give a little bit more details in the part dedicated to the methods.
Introduction:

Introduction:
-Line 60: distortion of time perception and speeded thoughts are two distinct/independent features according to the Greyson NDE scale, and speeded thoughts are not considered as a subtype of altered time perception.

- Line 107: there is a typo in the reference placed into brackets, it should read (Long & Holden, 2007). Besides, this reference appears twice in the references at the end of the manuscript.

Discussion:
-Line 238, it should read (Gallup, 1982).

-Line 242, authors write that NDEs are almost always associated with well-being, however, it would be interesting to mention that other studies have already identified negative NDEs, and sometimes in relatively large proportions. See, for example, the following articles:
Cassol, H., Martial, C., Annen, J., Martens, G., Charland-Verville, V., Majerus, S., & Laureys, S. (2019). A systematic analysis of distressing near-death experience accounts. Memory, 1–8.
Knoblauch, H., Schmied, I., & Schnettler, B. (2001). Different kinds of near-death experience: A report on a survey of near-death experiences in Germany. Journal of Near-Death Studies, 20(1), 15–29.
Lindley, J. H., Bryan, S., & Conley, B. (1981). Near-death experiences in a pacific northwest american population: The evergreen study. Journal of Near-Death Studies, 1, 104–124.

Besides, authors cite Charland-Verville et al. 2015 who report ‘less blissful’ NDEs. Charland-Verville et al. 2014 might be more appropriate to cite in this case.

-Line 245, still regarding the valence of NDEs, authors indicate that “experiences with the cut-off score of ≥7 GNDES points were reported significantly more often as pleasant (49%) than experiences with a lower score (13%)”, written this way this seems a bit misleading. Indeed, one could interpret that more intense NDEs (i.e. higher GNDES scores) are also more positive. But actually, this result might be biased by the focus of the GNDES on positive emotions.

-Lines 313-314: this entire sentence is misleading, “The central point in NDE research is that NDE do not only occur in healthy individuals but also during resuscitation.”. What do authors mean by healthy individuals? Do they refer to NDE-like (similar experience happening in absence of life threat)?

---

## Round 0.2 · accepted · Accept

The reviewers found the manuscript significantly improvement over the original submission and all criticisms were properly addressed.

Reviewer 1 ·

Basic reporting

The writing of this revision is clear and unambiguous. Literature background is very thorough. Article structure is professional. Results are relevant to the hypothesis.

Experimental design

This original research has a well-defined research question that explores a new potential predictor of near-death experiences, which may have implications for understanding their underlying physiological mechanisms. The investigation was performed ethically and the methods described in sufficient detail to permit replication.

Validity of the findings

As the authors suggested, their findings from this crowdsourced internet study should be replicated conceptually using other methods that do not rely on internet respondents. Underlying data are adequately provided and appear robust and statistically sound. Conclusions are well stated and limited to supporting findings. Speculation is now identified as such,

Additional comments

This revision is a significant improvement over the original submission, in that several points are now clarified that were ambiguous or insufficiently supported in the original.

Reviewer 2 ·

Basic reporting

no comment

Experimental design

Thanks to the modifications, the experimental design is now clearer.

Validity of the findings

no comment

Additional comments

The authors addressed the issues raised by the reviewers. The manuscript has been improved.

Reviewer 3 ·

Basic reporting

all comments have been addressed

Experimental design

no comment

Validity of the findings

no comment

Additional comments

all comments have been addressed